# Export of calcium carbonate corrosive waters from the East Siberian Sea

Leif G. Anderson[1], Jörgen Ek[2], Ylva Ericson[3], Christoph Humborg[2,4], Igor Semiletov[5,6,7], Marcus Sundbom[2], Adam Ulfsbo[1,8]

[1]Department of Marine Sciences, University of Gothenburg, 412 96 Gothenburg, Sweden
[2]Baltic Sea Centre, Stockholm University, 106 91 Stockholm, Sweden
[3]The University Centre in Svalbard, Pb 156, 9171 Longyearbyen, Norway
[4]Department of Environmental Science and Analytical Chemistry, Stockholm University, 106 91 Stockholm, Sweden
[5]International Arctic Research Center, University Alaska Fairbanks, Fairbanks, AK 99775, USA
[6]Pacific Oceanological Institute, Russian Academy of Sciences Far Eastern Branch, Vladivostok 690041, Russia
[7]The National Research Tomsk Polytechnic University, Tomsk, Russia
[8]Division of Earth and Ocean Sciences, Nicholas School of the Environment, Duke University, Durham, NC 27704, USA

*Correspondence to*: Leif G. Anderson (leif.anderson@marine.gu.se)

**Abstract.** The Siberian Shelf Seas are areas of extensive biogeochemical transformation of organic matter, both of marine and terrestrial origin. This in combination with brine production from sea ice formation results in a cold bottom water of relative high salinity and partial pressure of carbon dioxide (pCO2). Data from the SWERUS-C3 expedition compiled on the icebreaker Oden in July to September 2014 show the distribution of such waters at the outer shelf, as well as their export into the deep central Arctic basins. Very high pCO2 water, up to ~1000 µatm, was observed associated with high nutrients and low oxygen concentrations. Consequently, this water had low saturation state with respect to calcium carbonate down to less than 0.8 for calcite and 0.5 for aragonite. Waters undersaturated in aragonite were also observed in the surface in waters at equilibrium with atmospheric CO2, however, at these conditions the cause of under-saturation was low salinity from river runoff and/or sea ice melt. The calcium carbonate corrosive water was observed all along the continental margin and well out into the deep Makarov and Canada Basins at a depth from about 50 m depth in the west to about 150 m in the east. These waters of low aragonite saturation state are traced in historic data to the Canada Basin and in the waters flowing out of the Arctic Ocean north of Greenland and in the western Fram Strait, thus potentially impacting the marine life in the North Atlantic Ocean.

## 1 Introduction

The decrease of pH in the ocean as a result of uptake of anthropogenic carbon dioxide ($CO_2$) from the atmosphere is noted Ocean Acidification (e.g. Doney et al., 2009). The change in pH caused by the increase of partial pressure of $CO_2$ ($pCO_2$) from about 280 µatm in preindustrial times to about 400 µatm today is in the order of 0.1 pH unit. A continued usage of fossil fuels at the present rate has been projected to lead to an atmospheric level of close to 1000 µatm at the end of this

century, resulting in a further decrease of pH by about 0.3 units in oceanic surface waters (e.g. Orr et al., 2005). Changes in pH of this range will likely impact the marine ecosystem as well as society (e.g. Gattuso et al., 2015).

Biogeochemical processes in the Arctic Ocean can have a much greater impact on Ocean Acidification than that of anthropogenic $CO_2$ (Anderson et al., 2009; Anderson et al., 2011). Investigation of the East Siberian shelf area illustrates the substantial impact of organic matter decay on lowering pH and thus also the calcium carbonate saturation state (Semiletov et al., 2016). The corrosive shelf water spreads far out into the deep central Arctic Ocean at a depth centred around 150 m and potentially further into the North Atlantic Ocean (Azetsu-Scott et al., 2010). Here it can have negative effects on aragonite-shelled pelagic organisms like the pteropods (e.g. Comeau et al., 2010), which are important in the Arctic Ocean and North Atlantic food web.

The Arctic Ocean is one of the oceans that experience the lowest natural pH, a result of the high gas solubility in cold waters. On the other side the change in pH for a given increase in $pCO_2$ is less in cold waters than in warm, as the buffer capacity for $CO_2$ is less. However, there are several other processes than air-sea exchange of $CO_2$ that impact pH of Arctic waters. Some Arctic shelf seas, such as the Chukchi Sea and the Barents Sea, are among the most productive of the world oceans and as such experience large natural seasonal variability in $pCO_2$ and pH. One consequence of the high primary productivity is low $pCO_2$ in the inflow shelf seas (Carmack and Wassmann, 2006; Pipko et al., 2002) is that the surface waters of the central Arctic Ocean largely are undersaturated in $CO_2$ (Jutterström and Anderson, 2010). This is a result of the waters being covered by sea ice that hampers air-sea flux, before the atmospheric exchange compensates for the $CO_2$ taken up by the phytoplankton. Decreased summer sea ice cover in the central Arctic Ocean will likely reduce this effect and adjust the surface water closer to atmospheric equilibrium.

Increased $pCO_2$ levels decrease carbonate ion ($CO_3^{2-}$) concentrations, which also reduces the solubility of calcium carbonate crystals, illustrated by the reaction;

$$CaCO_3(s) \leftrightarrows Ca^{2+} + CO_3^{2-} \tag{1}$$

Where the solubility is determined by the chemical solubility product, $K_{so}$, which is the product of the calcium and carbonate ion concentrations at chemical equilibrium, i.e. $K_{so} = [Ca^{2+}]^{equilibrium} \times [CO_3^{2-}]^{equilibrium}$. In the ocean $K_{so}$ is dependent on temperature, salinity and pressure. To determine if the conditions are such that the water is supersaturated or undersaturated with respect to calcium carbonate one often uses the saturation state, $\Omega$.

$$\Omega = \frac{[Ca^{2+}]^{observed} \times [CO_3^{2-}]^{observed}}{K_{so}} \tag{2}$$

Hence, if $\Omega$ is less than 1 the water is undersaturated and if greater than 1 it is supersaturated.

Calcium carbonate crystals can have different structures, each with a specific $K_{so}$. The most common are calcite and aragonite, but in cold environments like sea ice also ikaite can form (e.g. Dieckmann et al., 2008). Calcite and aragonite are mainly formed by biological calcification where calcium carbonate is produced as part of an organism's structure. Even if

the crystals are built up of carbonate ions it is in most cases hydrogen carbonate ions ($HCO_3^-$) that are extracted by the organisms from seawater and converted to $CO_3^{2-}$ internally (e.g. Findlay et al., 2011). Thus there might not be a direct coupling between the biological formation rate of $CaCO_3$ minerals and ocean acidification. Instead it has been suggested that it is the ratio of $HCO_3^-$ to $H^+$ that controls the biotic $CaCO_3$ precipitation (Bach 2015), or more directly the increase in

$H^+$ (Cyronak et al., 2015). However, the opposite reaction, i.e. dissolution of $CaCO_3$ minerals, is influenced by ocean acidification through the chemical solubility product, at least as long as the crystals are directly exposed to seawater.

Among the biologically produced calcium carbonates, aragonite is more soluble than calcite making the former more sensitive to ocean acidification. During biological formation of calcite, magnesium can be incorporated instead of calcium, which increases the solubility; the higher the content of magnesium, the higher the solubility (e.g. Haese et al., 2014). High-

magnesium calcite has a higher solubility than aragonite making this the most sensitive form. A range of these mineral forms are produced by organisms living in the Arctic Ocean. Among those producing aragonite skeletons are cold-water corals and pteropodes, while most benthic organisms tend to produce a larger fraction of calcite. Consequently there is a substantial diversity in the biological produced $CaCO_3$ minerals and thus also different sensitivities to ocean acidification.

The very large shelf areas north of Siberia are heavily impacted by river input, adding fresh water and chemical constituents.

The direct addition of high $pCO_2$ runoff together with biochemical processes such as primary production and decay of organic matter, of both marine and terrestrial origin, strongly impact the calcium carbonate saturation state. The distribution of this specific water characteristic on the shelf as well as its export into, and circulation within, the deep basins is determined by the atmospheric pressure field that drives the oceanic currents (e.g. Jahn et al., 2010; Pipko et al., 2011).

In this contribution we show the importance of the Siberian shelves in producing calcium carbonate undersaturated waters

that spread out into the deep Arctic Ocean and further towards the Atlantic Ocean. The processes behind the under-saturation are elucidated as well as the flow path of this signature in the Arctic Ocean.

**2 Methods**

This contribution is based on data that were collected during the SWERUS-C3 (Swedish – Russian – US Arctic Ocean Investigation of Climate-Cryosphere-Carbon Interactions) expedition with the Swedish icebreaker *Oden* along the outer

Siberian shelf and covers a longitudinal range between about 125 $^o$E to 175 $^o$W (see Figure 1 for station positions). Sampling was intensified across the shelf break along sections (A-F). Furthermore, historic data are used to evaluate how the waters from the Siberian shelf are spread out in the central Arctic Ocean and further towards Fram Strait on their way to the North Atlantic Ocean. These include data from north of Greenland collected during the Oden 1991 cruise (Anderson et al., 1994), data from the northern Fram strait collected in 2002 (Jutterström et al., 2008), and data from the Canada Basin

collected in 2005 (Jones et al., 2008a).

The SWERUS-C3 data were gathered at about 100 stations, occupied between 15 July and 25 September 2014. Water samples were collected using a rosette system equipped with 24 bottles of Niskin type, each having a volume of 7 L. The

bottles were closed during the return of the CTD-rosette package from the bottom to the surface and water samples for all constituents were drawn soon after the rosette was secured in the CTD container.

The following constituents are used; dissolved inorganic carbon (DIC) (only stations visited after 24 August), total alkalinity (TA), pH, oxygen, nutrients, and bottle salinity. The order of sampling was determined by the risk of contamination meaning that oxygen samples were collected first followed by the carbonate system parameters, the nutrients, and salinity. Water samples for salinity were analysed for more than 90% of the depths and when no data were available the CTD salinity was used. Temperature data were taken from the CTD.

Salinity and temperature data were collected using a SeaBird 911+ CTD with dual SeaBird temperature (SBE 3), conductivity (SBE 04C) and oxygen sensors (SBE 43) attached to the rosette system. Salinity was calibrated against deep water samples analysed on board using an AUTOSAL 8400B lab-salinometer. The salinometer was calibrated using one standard sea water ampoule (IAPSO standard sea water, OSIL Environmental Instruments and Systems) before each batch of 24 samples. The accuracy of the Autosal salinities and CTD salinities were both within ±0.003 psu and the accuracy for temperature ±0.002 $^{\circ}$C.

DIC was determined by a coulometric titration method based on Johnson et al. (1987), having a precision of 2 $\mu$mol kg$^{-1}$, estimated from duplicates, with the accuracy set by calibration against certified reference materials (CRM), supplied by A. Dickson, Scripps Institution of Oceanography (USA). TA was determined by automated open-cell potentiometric titration (Haraldsson et al., 1997), precision better than 2 $\mu$mol kg$^{-1}$, with the accuracy set the same way as for DIC. Seawater pH was determined by a spectrophotometric method, based on the absorption ratio of the sulphonephtalein dye, $m$-Cresol Purple (mCP) (Clayton and Byrne, 1993). Purified mCP (Liu et al., 2011) was purchased from the laboratory of Robert H. Byrne, University of South Florida, USA. The accuracy was estimated to 0.006 from internal consistency calculations of analyzed CRM samples and the precision, defined as the absolute mean difference of duplicate samples, was ~0.001 pH units.

Oxygen was measured by an automated Winkler titration system giving a precision of ~1 $\mu$mol kg$^{-1}$. The accuracy was set by titrating known amounts of $KIO_3$ salts that were dissolved in sulphuric acid. As the amount was known to better than 0.1% the accuracy should be significantly better than the precision. The deviation from saturation concentration was computed as apparent oxygen utilization (AOU) as $[O_2]^{sat}$ - $[O_2]^{measured}$.

Dissolved inorganic nitrogen species ($NH_4$, $NO_2$+$NO_3$), orthophosphate ($PO_4$) and dissolved silicate were measured on board using a four-channel continuous flow analyser (QuAAtro system, SEAL Analytical). Within each analysis run a calibration was done using standard solutions of $NH_4Cl$, $KNO_3$, $K_2HPO_4$ and a commercial stable silica-compound solution. Analysis quality was further assured by automatic drift control using standard solutions and including CRM solutions prepared from commercial ampoules (QC RW1, Batch VKI-9-3-0702) in the analysis run. Accuracy expressed as the median deviation of the measurements from the CRM solution for $NH_4$, $NO_2$+$NO_3$ and $PO_4$ was -0.15, -0.11, and -0.018 $\mu$mol L$^{-1}$, respectively. The CRM is not certified for silicate. The precision was, based on 28 determinations of standards, 3.7%, 1.2%, 2.7% and 1.3% for $NH_4$, $NO_2$+$NO_3$, $PO_4$ and $SiO_4$, respectively.

The saturation states of the two major forms of calcium carbonate ($\Omega^{aragonite}$ and $\Omega^{calcite}$) and $pCO_2$ were calculated from the combination of pH and TA, as well as pH and DIC when the latter was available, using CO2SYS (van Heuven et al., 2011) using the carbonate dissociation constants ($K_1$ and $K_2$) of Lueker et al. (2000), the solubility product, $K_{so}$ according to Mucci, (1983) and the salinity-calcium ion concentration ratio of Riley and Tongudai (1967). When two computations were performed the reported values are the average of the two calculated for each sample. Input data included salinity, temperature, $PO_4$ and $SiO_4$ data. The uncertainty was computed using a Monte Carlo approach (Legge et al., 2015) and expressed as double standard deviation was about 2.5% for $pCO_2$ and less than 1% for $\Omega$.

Furthermore, as the carbonate system was over-determined the internal consistency (or thermodynamic consistency) was assessed by comparing measured values to calculated values (from any two of the three determined parameters DIC, TA and pH) using the same CO2SYS Matlab program (Van Heuven et al., 2011). The average mean differences between measured and calculated values were evaluated for pH to about 0.02 and for TA and DIC to about 7 μmol/kg for each.

## 3 Results

Water undersaturated with respect to calcium carbonate was observed along the bottom of the shelf at most stations (Figure 2), associated with high nutrient and low oxygen concentrations. The strongest signal was north of the New Siberian Islands, a local minimum in saturation state at about 157 °E followed by a less strong signal further to the east. Low saturation state with under-saturation in aragonite was also observed in the surface water north of the New Siberian Islands. This water had low salinity, high phosphate, as well as high silicate concentrations and relatively high $pCO_2$ despite having nitrate concentrations as low as about 0.2 μmol/L.

The extension of this high $pCO_2$ and low $\Omega^{aragonite}$ water out into the deep basin was seen in a layer centred at about 100 m depth all along the shelf break from the east of the Lomonosov Ridge to 180 °E (Figure 3). Over the Lomonosov Ridge (section A) there was a faint minimum in $\Omega^{aragonite}$ even if the waters never were undersaturated. Moving east to section B undersaturated waters were observed both in the surface and at approximately 50 m depth. Further to the east, sections C, D and E, have strong $\Omega^{aragonite}$ minima centred at ~100 m depth. In the most eastern section of this study, section F, the minimum was less intense and was also found at somewhat greater depth, ~150 m. In all sections the minimum in $\Omega^{aragonite}$ was closely followed by a maximum in $pCO_2$ (Figure 3). In section C and even more so in D and E the $\Omega^{aragonite}$ minima were observed in close vicinity to the sediment.

This low saturated water has been observed in the Canada Basin on several occasions (e.g. Yamamoto-Kawai et al., 2009; Anderson et al., 2010). During the Beringia 2005 expedition where $\Omega^{aragonite}$ levels as low as 0.6 were observed at 100 m depth (Figure 4), the layer of under-saturation was between 50 and 100 m thick and had the upper limit at about 75 m depth. It also had the typical signature of low oxygen and high nutrient concentrations from decay of organic matter. Water with high silicate and undersaturated in aragonite was observed at the tip of the Morris Jesup Plateau in 1991 (Figure 5) where the subsurface waters from the Canadian Basin follows the continental margin towards the Fram Strait (Rudels et al., 1994).

Water of similar signature, but less pronounced, was also seen at depth of 50-75 m in the north western Fram Strait in May 2002, while the overlying low saline surface water was well above saturation with respect to calcium carbonate (Figure 6). A distinct maximum in silicate was present at the Greenland continental margin and centred around a salinity of 33. It was associated with a minimum in pH and thus also in $\Omega^{aragonite}$, but with a less pronounced oxygen minimum (AOU maximum)

**5 Discussion**

The two major factors that impact calcium carbonate solubility in the surface ocean are salinity and pH, where the former effects the calcium ion and DIC concentrations and the latter controls the relative distribution of the inorganic carbon speciation. In the open ocean the latter is mainly determined by $CO_2$, either controlled by exchange with the atmosphere or through primary production / decay of organic matter. The concentrations of nutrients and AOU are high close to the bottom

at most stations occupied on the shelf (Fig. 2), showing the importance of organic matter decay in controlling the chemical signature of the Siberian shelf bottom waters. Low $\Omega^{aragonite}$ and $\Omega^{calcite}$ values are associated with high $pCO_2$ (low pH) and high nutrient concentrations as well as low oxygen (high AOU). The corrosive water extending out into the deep basin is strongly associated with high $pCO_2$ (Figure 3) and high nutrients, further illustrating that decay of organic matter also here is the cause of this signature.

However, mixing with low salinity water also has an impact, especially on the shelf where the influence from river discharge is stronger and melting sea ice occurs that add to lowering salinity. As illustrated in Figure 7a aragonite under-saturation is found in most surface waters of salinity < 28, waters that are found north of the New Siberian Islands (Figure 2). In this region the surface water (top 10 m) nitrate concentration is very low, average 0.25±0.05 µmol/L, illustrating that phytoplankton blooms are well advanced and that nitrate is the limiting nutrient as the phosphate concentration is high,

average 1.1 ±0.1 µmol/L (Figure 2). Despite these signatures of primary production, $CO_2$ is slightly supersaturated, average 420 ±30 µatm and oxygen is close to equilibrium, average 100±1 %, i.e. AOU is around zero. Hence the rate of $CO_2$ consumption by primary production is exceeded by respiration of organic matter low in nutrient content, and this is likely of terrestrial origin (Anderson et al., 2009). $CO_2$ can also be produced by photo degradation of dissolved organic matter (e.g. Cory et al., 2015), at least if the water is not too turbid. However, our data cannot assess the relative roles of these two

processes.

The very high silicate concentrations indicate that mainly river runoff and not sea-ice melt is the source of freshwater to low salinity waters. Consequently, this region is most likely heavily impacted by the plume of the large Siberian Rivers, here dominated by the Lena River. The fact that $CO_2$ is supersaturated strongly indicates that the river runoff do not only lower the calcium and carbonate concentrations by dilution, but also decreases pH by addition of $CO_2$. This could be either from

high $pCO_2$ in the river water itself or by decay of organic carbon that is added by the river runoff. Considering a freshwater residence time on the Laptev shelf of several years (Schlosser et al., 1994) the latter is the most plausible explanation as $CO_2$ from the runoff would have time to largely equilibrate during the transit to the area of observation. This is supported by reconstructed velocity fields (using a 4D variational approach) and trajectories of passive tracers launched in the East

Siberian Arctic Shelf which show that during 2 years the surface water mass approaches the area near the North Pole (Shakhova et al., 2015).

Summer water $pCO_2$ values of the central Arctic Ocean has normally been observed to be well below atmospheric levels (e.g. Jutterström and Anderson, 2010; Ulfsbo et al., 2014), but during the Russian North Pole-33 drifting station $CO_2$ supersaturation of up to 100 µatm was observed over the Amundsen basin at about 86 °N (Semiletov et al., 2007). This is the region where the Lena river plume typically passes (e.g. Anderson et al., 2004) and these high $pCO_2$ values can thus be an indication of; a direct contribution from the Lena River plume, mineralization of terrestrial DOC following the river plume, rapid turnover of sub-ice DOM which is closely connected with high sea-ice production, or any combination of these sources (Semiletov et al., 2007). A time lag of 4-6 years between the Lena River discharge anomalies, ice conditions in the Siberian seas, and ice export through Fram Strait was found with a correlation of ~0.7 (Semiletov et al., 2000). During this passage over the central Arctic Ocean the $pCO_2$ levels decreases by; outgassing, mixing of water masses as well as by primary production, resulting in that the surface water that flow out through Fram Strait in the East Greenland Current is supersaturated with respect to calcium carbonate (Figure 6).

The dilution is not a dominating process of low $\Omega^{aragonite}$ throughout the water column in the study region as seen in a DIC – TA plot (Figure 7b). The degree of saturation varies more with shifts in DIC by air-sea exchange (direction of arrow 1) and primary production / organic matter decay (direction of arrow 3) than dilution (direction of arrow 2), where the latter has an effect on both TA and DIC. No clear trend can be seen when $\Omega^{aragonite}$ is plotted as a function of salinity (Figure 7c) and the data certainly do not follow the dilution line (representing dilution of Atlantic Layer Water having S = 34.85, T = -1 °C, DIC = 2170 µmol/kg, TA = 2300 µmol/kg with water of zero S, DIC and TA), even if there are less water supersaturated at the lower salinities. Rather, the low $\Omega^{aragonite}$ is associated with high $PO_4$ implying organic matter decay as the main cause. This conclusion is strengthened by the association between $\Omega^{aragonite}$ and AOU (Figure 7d). A line is drawn that represents a typical surface water (S = 33, T= -1 °C, $PO_4$ = 1 µmol/L, $NO_3$ = 16 µmol/L, AOU = 0, DIC = 2046 µmol/kg, TA = 2169 µmol/kg) that is impacted by primary production and decay of organic matter having the classical RKR ratio of the elements $C:N:P:O_2$ = 106:1:16:-138 (Redfield et al., 1963). This computation is done assuming a classical marine organic matter having this elemental composition. However, the correlation between $\Omega^{aragonite}$ and AOU will not be much impacted if instead terrestrial organic matter of lower nutrient content decays. Most of the data, in general, follow the shape of this line even if the spread is substantial. It should be noted that all data collected during the expedition is included, i.e. including water originating from both the Atlantic and the Pacific Oceans with large variability in time since contact with the atmosphere. One example is the blue data that deviates from the line in the upper left of the graph, which likely is a result of photosynthesis in the surface water where outgassing of oxygen to the atmosphere is faster than the uptake of $CO_2$. These are surface waters of salinities around 32 (Figure 7a & c).

The low $\Omega^{aragonite}$ water observed along the continental margin (Figure 3) likely flows to the east and spreads out into the deep basin. In 2005 it was seen in the Canada Basin (Figure 4) and likely confined to the Beaufort Gyre and hence its

extension depends on the atmospheric pressure field (e.g. Proshutinsky et al., 2009; Carmack et al., 2016). Decadal variability in the fresh water content of the Beaufort Gyre experienced a significant change in the 1990s, forced by a change of the atmospheric circulation regime. For instance, in 1979 silicate concentrations as high as 40 µmol/L were observed over the Lomonosov ridge close to the North Pole during the LOREX ice camp (Moore et al., 1983), but in 1991 the silicate

maximum was completely absent even in the Makarov Basin close to the North Pole (Anderson et al., 1994). One consequence of this variable extent of the Beaufort Gyre is that the export of the high nutrient, low $\Omega^{aragonite}$ water out into the North Atlantic Ocean is fluctuating with time.

Jones et al. (2003) identified Pacific-originating water by its excess phosphate concentration relative to nitrate and used the nitrate to phosphate relationship to trace it in the North Atlantic Ocean and found significant fractions in the Fram Strait, the

10 Denmark Strait, and the Labrador Sea, but variable between different years. For many of the cruises discussed by Jones et al. (2003) there were no carbonate system data measured and hence $\Omega^{aragonite}$ could not be computed. However, it is evident that the aragonite undersaturated water is well confined to the water high in nutrients and with an excess of phosphate (Figure 8), i.e. a property that was observed in parts of the North Atlantic by Jones et al. (2003). Hence, this undersaturated water is also likely exported well into the North Atlantic Ocean.

The 2002 observations show a minimum of less than 1.4 in aragonite saturation, but not undersaturated, in the north western Fram Strait (Figure 6). This minimum is centred around a salinity of 33, but the computed fraction of Pacific Water was about 2% with the major fraction of freshwater being river runoff (Jones et al., 2008b). Hence, the signature of calcium carbonate undersaturation was lessened from the source to the observation of this investigation, i.e. the East Siberian Arctic Shelves, by water mass mixing.

However, when one compares the conditions of the waters flowing into the Arctic Ocean in the West Spitsbergen Current and the one flowing out in the East Greenland Current there are distinct differences (Figure 9). The silicate concentration is at least 2 µmol/l higher in the top 200 m while the $\Omega^{aragonite}$ values are in the order 0.2 lower in the top 100 m. This is most likely a result of addition of freshwater by both river runoff and sea ice melt, but even more important is the decay of organic matter. The effect by freshening can decrease $\Omega^{aragonite}$ by no more than 0.05 under the conditions observed, pointing to the

importance of organic matter decay during the circulation of these waters in the Arctic Ocean.

## 6 Conclusions

Corrosive surface water is exported to the central Arctic Ocean by the plumes of the large Siberian Rivers, dominated by the Lena River in the Laptev – East Siberian Seas region. This signature is maintained in the shelf by photo degradation (e.g. Cory et al., 2015) and/or microbial degradation of organic matter that to a large degree is of terrestrial origin (e.g. Alling et

al., 2012; Tesi et al., 2014; Vonk et al., 2014). However, the signature of surface water undersaturated with respect to calcium carbonate does not spread far out into the deep central basins as mixing with surrounding water masses and

outgassing of $CO_2$ to the atmosphere increase the saturation state with time. Furthermore, the saturation state is also lowered by primary production in the surface waters when light and nutrients are available.

Highly calcium carbonate corrosive waters are produced along the bottoms of the East Siberian Seas through degradation of organic matter. This water is exported into the deep Makarov and Canada Basins at a depth range of about 50-150 metres.

The signature of this subsurface water is maintained within the Beaufort Gyre and is exported out to the North Atlantic through eastern Fram Strait, and likely also through the Canadian Arctic Archipelago.

The salinity range of this water is 32-34 and there is a potential for this water to be mixed up to layers inhabited by aragonite forming organisms such as pteropods. Such mixing can be either a result of wind or through sea ice formation when the surface layer gets homogenized by brine release, or when it enters the East Greenland Current.

**Acknowledgments**

This research was supported by the Swedish Knut and Alice Wallenberg Foundation (KAW); the Swedish Research Council, VR (L.G.A. 621-2013-5105); the Swedish Research Council Formas (A.U. #2014-1165), the Russian Government for support (I.S. grant no. 14,Z50.31.0012/03.19.2014). We thank the supporting crew and Captain of I/B Oden, the support of the Swedish Polar Secretariat, as well as the constructive comments by Johan Ingri and two anonymous reviewers on an 15  earlier version of this manuscript.

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

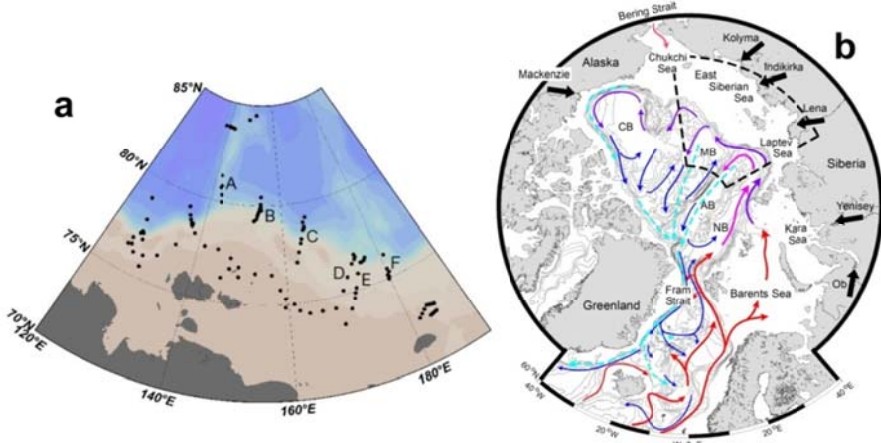

**Figure 1. Map illustrating the station locations, with (a) sections noted and (b) general water mass circulation in the deep Arctic Ocean and its connection to surrounding oceans. The location of map in (a) is shown by the dotted insert in map (b).**

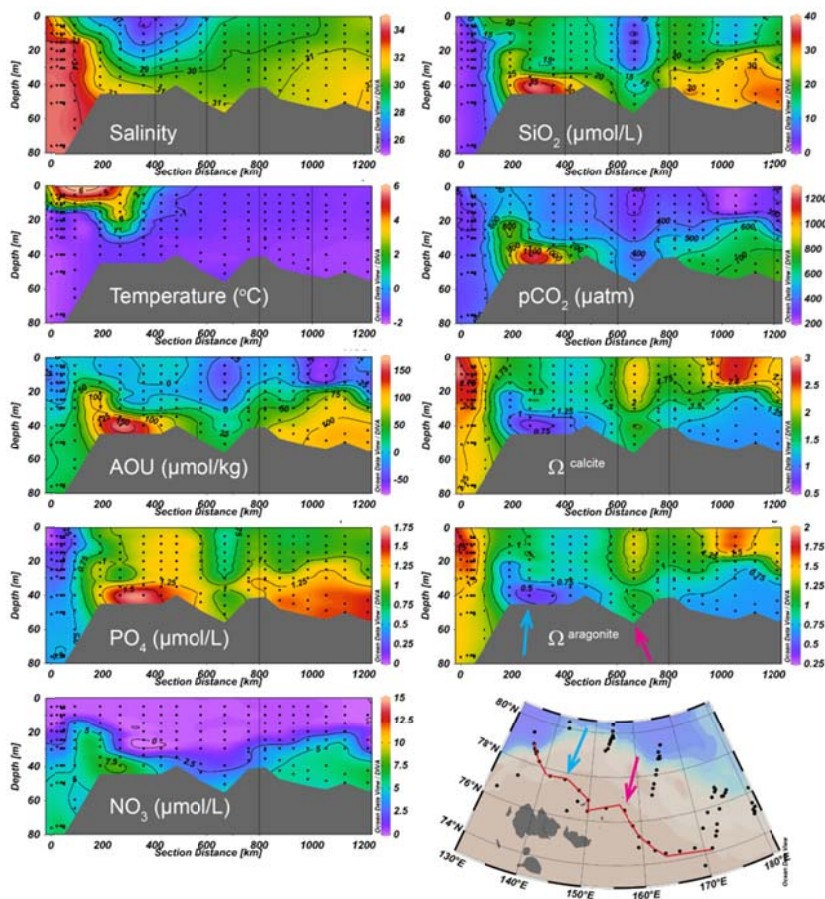

5   **Figure 2. Sections along the outer East Siberian continental shelf as shown by the red line of the map. The location of the minimum in saturation state of calcium carbonate is noted by the blue arrow and the local maximum by the purple arrow.**

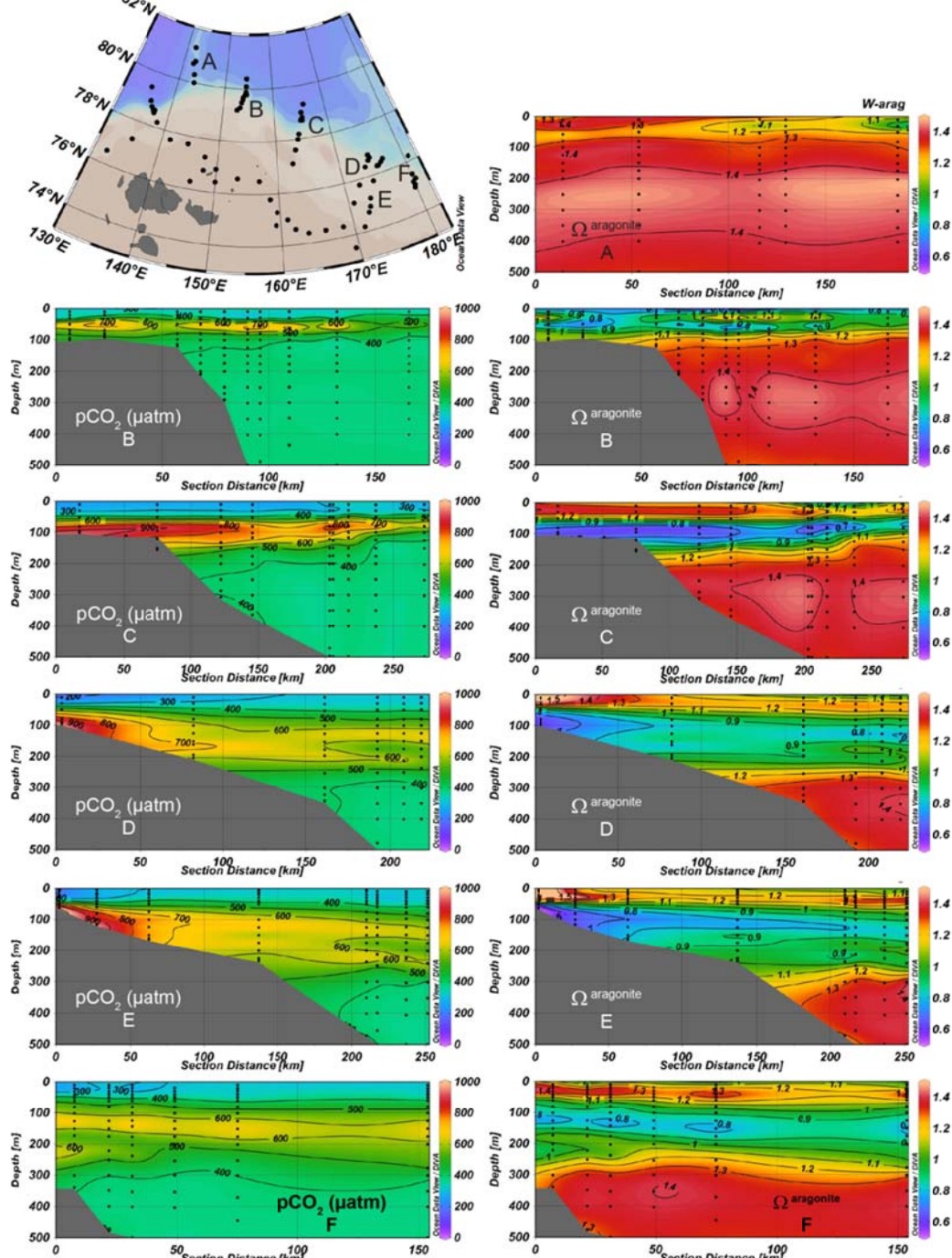

**Figure 3. Sections A to F across the East Siberian shelf break, partial pressure of carbon dioxide (pCO₂) to the left and saturation state of aragonite (Ω^aragonite) to the right. On all sections the colour scale is from 0 to 1000 µatm for pCO₂ and 0.5 to 1.5 for Ω^aragonite. Note the variable horizontal extent of the different sections. The locations of the sections are noted on the inserted map.**

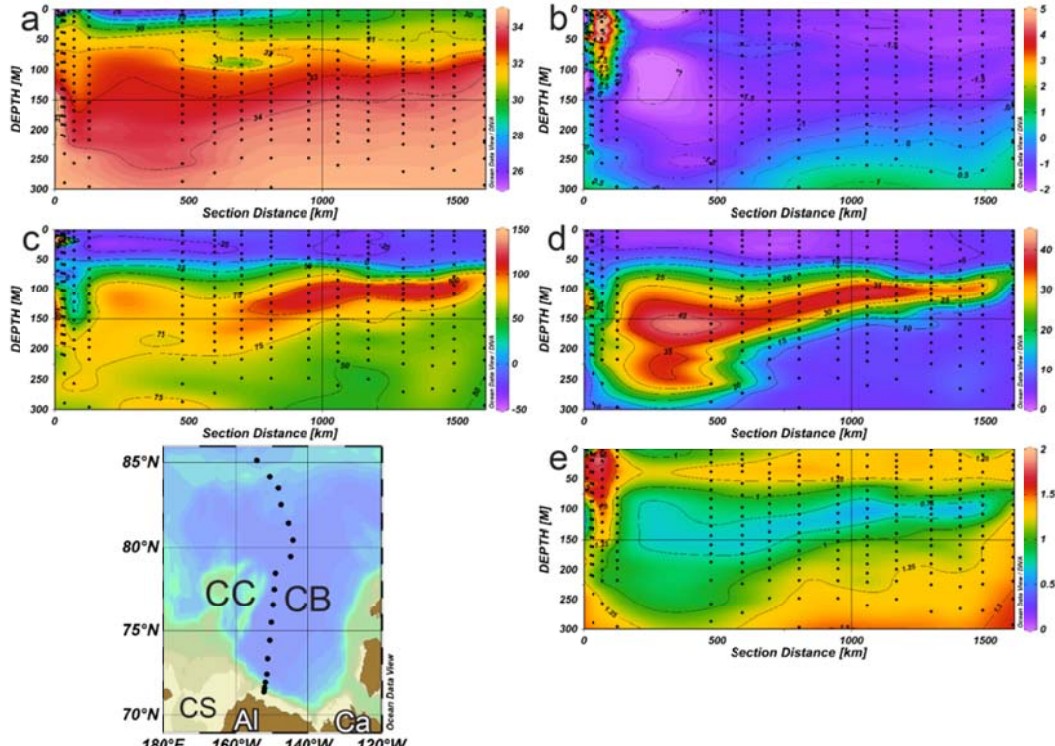

**Figure 4. Sections of (a) salinity, (b) temperature, (c) AOU, (d) silicate, and (e) $\Omega^{aragonite}$ in the top 300 m across the Canada Basin as observed during the Beringia 2005 expedition. Station positions are noted on the map, with the abbreviations being; CB = Canadian Basin, CC = Chukchi Cape, CS = Chukchi Sea, Al = Alaska, and Ca = Canada.**

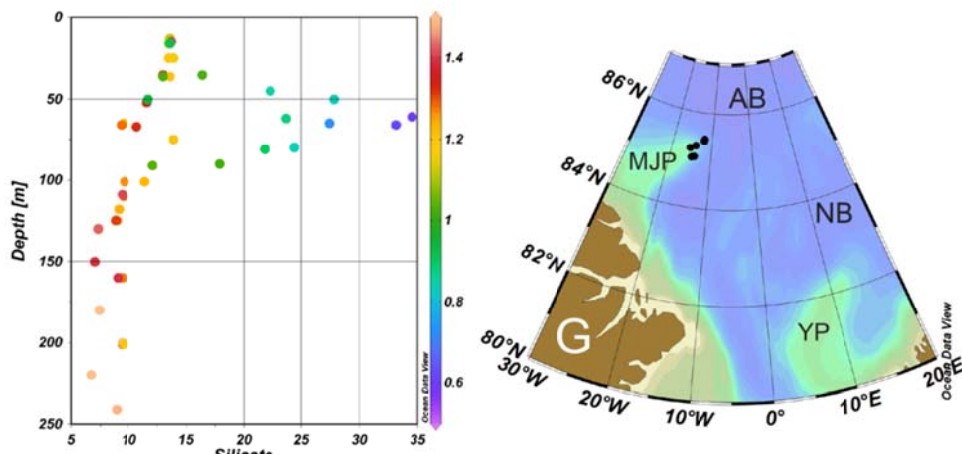

**Figure 5. Silicate profiles of six stations over the Morris Jesup Plateau (MJP), colour coded by $\Omega^{aragonite}$, as observed during the Oden 91 cruise. Station positions are noted on the map, with the abbreviations being; AB = Amundsen Basin, NB = Nansen Basin, YP = Yermak Plateau, and G = Greenland.**

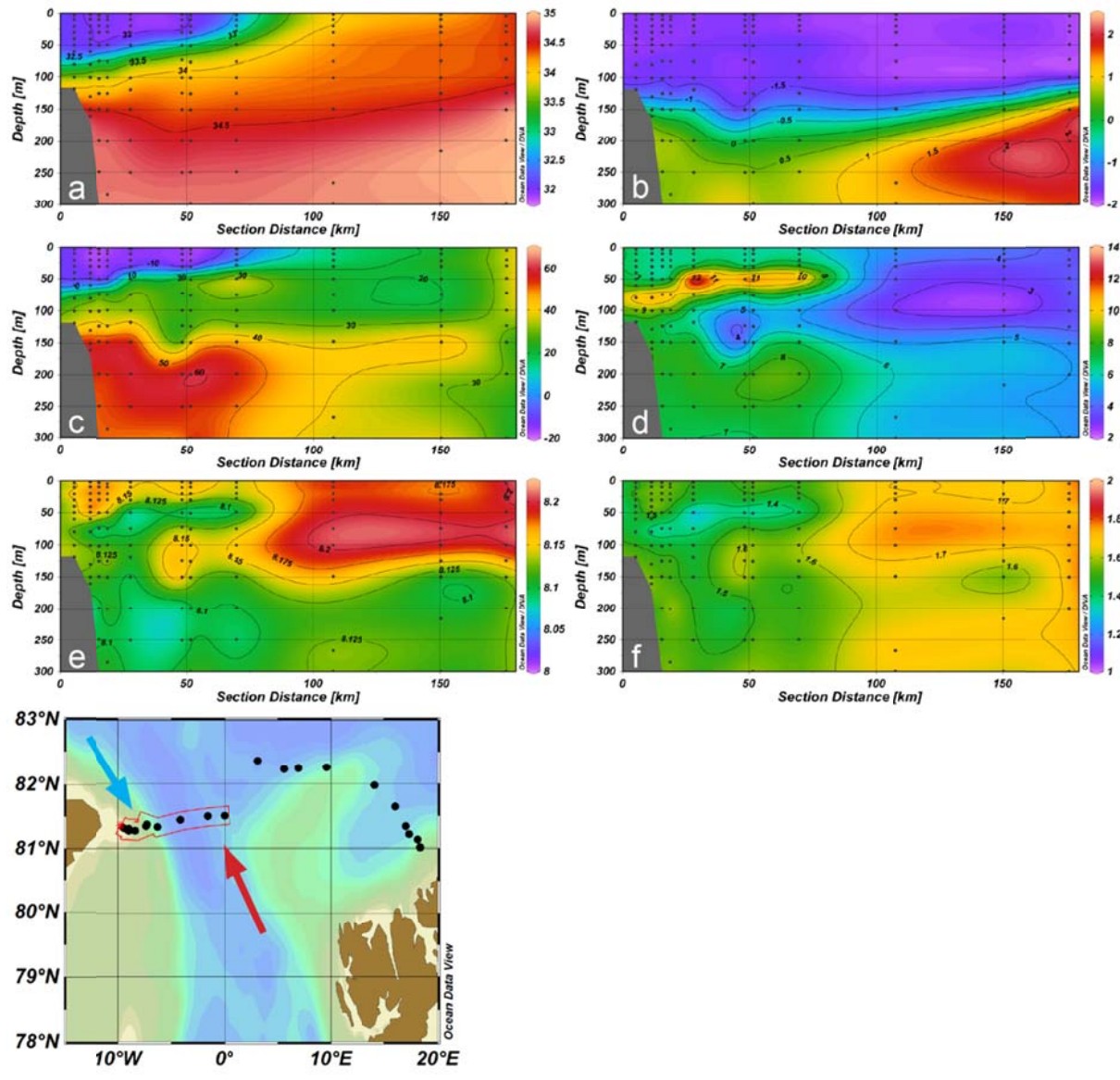

**Figure 6.** Sections of (a) salinity, (b) temperature, (c) AOU, (d) SiO$_2$, (e) pH$^{tot}_{in\ situ}$, and (f) $\Omega^{aragonite}$, across the northen Fram Strait at about 81.5 °N from the Yermak Plateau to Greenland as noted on the map, collected in May 2002. The arrows illustrate the north flowing warm Atlantic and the southflowing cold Arctic surface waters.

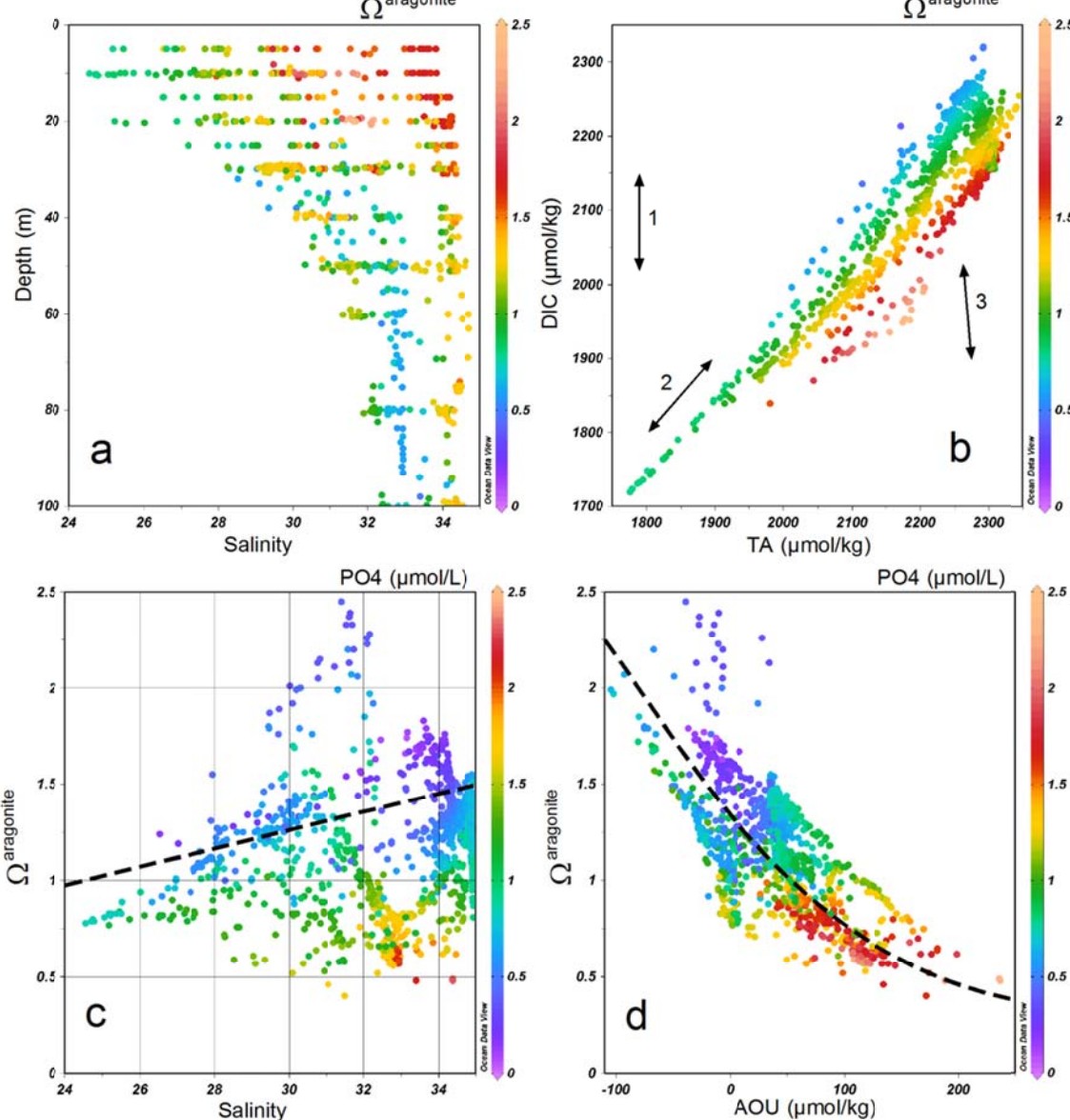

**Figure 7.** Plot of a) salinity versus depth b) DIC versus TA, both colour coded by $\Omega^{aragonite}$, c) $\Omega^{aragonite}$ versus salinity and d) $\Omega^{aragonite}$ versus AOU, the latter both colour coded by $PO_4$. In b) arrows are drawn where the slope are to illustrate the impact by air-sea $CO_2$ gas exchange (1), dilution of water with zero DIC and TA (2) and primary production (down) and decay of organic matter (3). The location and length of the arrows are arbitrary. In c) a line is drawn that represents the dilution of Atlantic Layer Water with water of zero S, DIC and TA, and in d) a line is drawn that represents a typical surface water that is impacted by primary production and decay of organic matter. Details of these lines are given in the text.

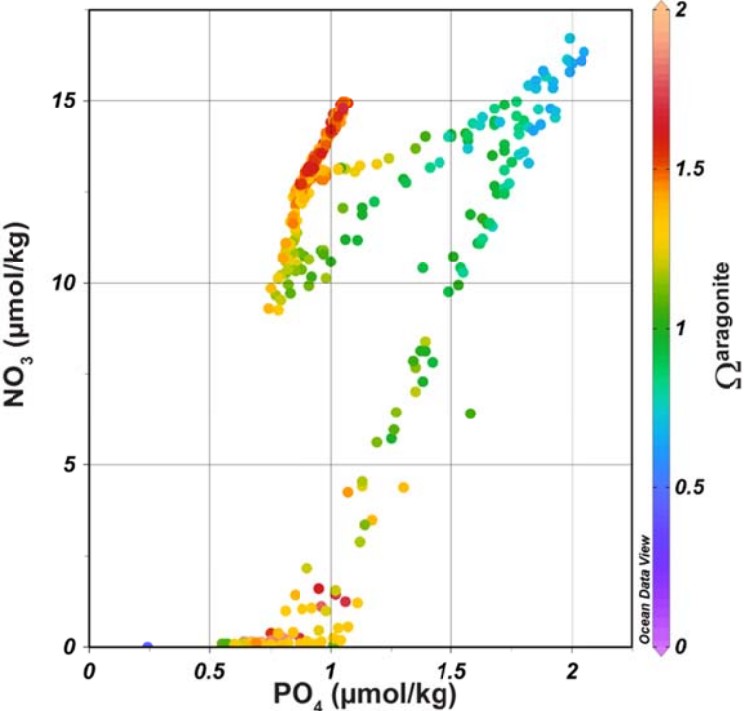

**Figure 8.** Nitrate versus phosphate as observed in the Canada Basin during the Beringia 2005 expedition (station locations same as in Fig. 4), colour coded by $\Omega^{aragonite}$.

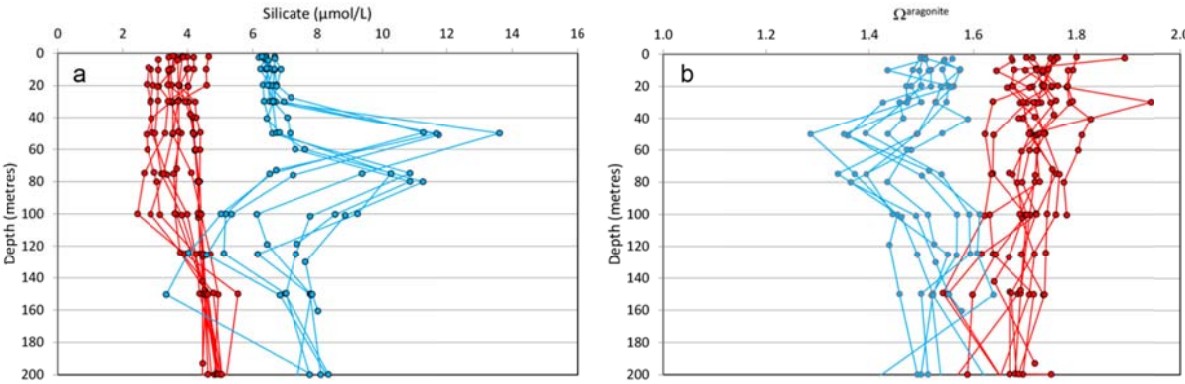

5  **Figure 9.** Depth profiles of silicate (a) and $\Omega^{aragonite}$ (b) in the Fram Strait around 82⁰ N in 2002. The 2002 are all computed from pH and TA and covers the whole strait. In the latter one can notice the clear division of the north flowing Atlantic water (red with a surface salinity around 34.5) and the south flowing Arctic Waters (blue with a surface salinity around 32). Both in and outflowing water has a surface temperature below -1 ⁰C making the effects of salinity and temperature on $\Omega^{aragonite}$ in the order of maximum 0.05.