# Peer review of "Export of calcium carbonate corrosive waters from the East Siberian Sea"

_Biogeosciences, 2016_

## Short Comment (SC1) · 16 Nov 2016

Leif and co-authors.

It would be very helpful for the readers if earlier work by Leif and Igor (e.g., Anderson et al., 2010; Semiletov et al., 2004; 2007) were included as these previous studies illustrate the emplacement of high pCO2/low pH waters in the East Siberian Sea (and by inference low calcium carbonate mineral saturation states). I think these earlier works should be included as context for the important ESS shelf-basin sections reported and discussed in this paper.

I agree with the authors about the importance of finding the export of highly corrosive waters from the East Siberian Sea shelf into thermocline waters of the Makarov and Canada basins. For the interest of the authors, there is also export

of high/pCO2/low pH/low saturation state water from the East Siberian Sea shelf through Long Strait (with the Siberian Sea Current) into the Chukchi Sea, which in turn will contribute to preconditioning of the Chukchi Sea and subsequent far-field effects. This was from the RUSALCA program and although I hesitate to self-cite, this finding is shown in the following paper: Bates, N.R., 2015. Assessing ocean acidification variability in the Pacific–Arctic Region (PAR) as part of the Russian–American Long–term Census of the Arctic (RUSALCA). Oceanography, 28(3), 36–45, http://dx.doi.org/10.5670/oceanog.2015.56.

---

## Referee Comment (RC1) · Anonymous Referee #1 · 12 Dec 2016

**Review on Export of calcium carbonate corrosive waters from the East Siberian Sea by Leif Anderson et al.**

Anderson et al. report on recent observations of various water masses in the Arctic Ocean from the SWERUS-C3 expedition onboard the icebreaker *Oden* in July to September 2014. The nutrient and carbonate system measurements are a valid addition to the still relative scarce data set from the Arctic Ocean. The data are well presented and discussed. The paper is very well written. I have a few minor comments and recommend publication in Biogeosciences.

p. 1, lines 31-32: for the importance of pteropods you might cite, for example, Comeau et al. (2010)

p. 2, lines 27-30: 'Even if the crystals are built up of carbonate ions it is in most cases hydrogen carbonate ions ($HCO_3^-$) that are extracted by the organisms from seawater and converted to $CO_3^{2-}$ internally (e.g. Findlay et al., 2011). Thus there might not be a direct coupling between the biological formation rate of $CaCO_3$ minerals and ocean acidification.' This topic has been discussed recently by Cyronak et al. (2015) and Bach (1915).

p. 3 lines 3-4: You might cite the paper on the solubility product of high-Mg calcite by Haese et al. (2014)

p. 3 lines 24-25: by measuring DIC, TA, and pH the carbonate system has been overdetermined (only stations visited after 24 August). Did you investigate the consistency of these data? (compare the recent discussion on inconsistencies in the marine carbonate system by Hoppe et al., 2012).

p. 5. 'The two major factors that impact calcium carbonate solubility in the surface ocean are salinity and pH.' The direct effect of salinity on calcium carbonate solubility is rather low, however, salinity is a proxy for total alkalinity which, together with another carbonate system parameter as for example pH, determines the saturation level.

**References**

[1] Bach, L.T. Reconsidering the role of carbonate ion concentration in calcification by marine organisms. *Biogeosciences*, 12(16):4939–4951, 2015.

[2] Comeau, S., R. Jeffree, J.-L. Teyssié, and J.-P. Gattuso. Response of the Arctic pteropod Limacina helicina to projected future environmental conditions. *PloS one*, 5(6):e11362, 2010.

[3] Cyronak, T., K.G. Schulz, and P.L. Jokiel. The Omega myth: what really drives lower calcification rates in an acidifying ocean. *ICES Journal of Marine Science: Journal du Conseil*, page fsv075, 2015.

[4] Haese, RR and Smith, J and Weber, R and Trafford, J. High-magnesium calcite dissolution in tropical continental shelf sediments controlled by ocean acidification. *Environmental science & technology*, 48(15):8522–8528, 2014.

[5] Hoppe, C.J. M., G. Langer, S.D. Rokitta, D.A. Wolf-Gladrow, and B. Rost. Implications of observed inconsistencies in carbonate chemistry measurements for ocean acidification studies. *Biogeosciences*, 9:2401–2405, 2012.

---

## Referee Comment (RC2) · Dr. ingri (Referee) · 18 Jan 2017

This is an interesting paper discussing corrosive waters from the East Siberian Sea. It is easy to follow the text but there are some parts in the text that needs clarification. The study should be published after minor revison (see below)

Comments

(text from introduction) High corrosive shelf water spreads far out into the deep central Arctic Ocean. Is that bottom waters or surface water or both?

(Discussion) how can calcium carbonate solubility be regulated by salinity? This must be explained. Salinity just reflects Ca2+ and CO32- concentrations, or?

The conclusion section must be rewritten and text added to the discussion (see below).

Which data in the study support the staterment "This signature is maintained in the shelf by microbial degradation of organic matter that to a large degree is of terrestrial origin"? Photodegradation of dissolved organic matter has been shown to be a significant or even dominant mechanism of oxidation of this material in sunlit waters. In shallow arctic lakes and streams the photo-defgradation of DOM can greatly exceed bacterial respiration (see discussion in Cory et al, 2015 Biogeosciences 12, 6669-6685). Hence, the authors must shortly discuss the role, influence, of photo-degradation (if any) for the formation of corrosive surface waters.

Furthermore, the statement "Mixing and uptake of $CO_2$ from the atmosphere prevent the calcium carbonate undersaturated water to spread out far from the continental margin" This statement is unclear. Which undersaturated water? Which type of mixing? Uptake of $CO_2$?

―――――――――――――――――――

---

## Referee Comment (RC3) · Anonymous Referee #3 · 20 Jan 2017

This short manuscript reports on "carbonate corrosive waters" in the Arctic Ocean and conclude that respiration of terrigenic organic matter is in part responsible for their formation.

The results arise from an extensive field campaign, the methods appear sound and the results support the conclusions. Many of the statements, however, are not substantiated by either references or clear explanations in the text. I understand that this manuscript is aiming for a focussed special issue, but BG is a multidisciplinary journal and the article should stand on its one. The opening statement on p. 1 l. 27, for instance, may be general knowledge in the field, but should nevertheless be supported by references for the readers to examine. My second general comment is with respect to some shakiness of the chemical nomenclature used. I give specific examples relative to these issues below.

Specific comments:

-I feel that the 1st paragraph of the introduction should be switched with the 2nd. This way, the text would first describe and define ocean acidification. Then the controls by biogeochemical process es vs anthropogenic CO2 dissolution would be contrasted.

-Equations 1–2 are unclear as to which concentration refer those at equilibrium with the solid and t to those measured. Specifically: Line 20: Eq. 2 and the definition of KSo refer to the same quantities in the text, while they should be different. KSo should be the product of the calcium and carbonate ion activity *at equilibrium*. Then, the ion activity product (IAP) refers to those in the sample.

-$\Omega$ is the saturation state, not the solubility state. Elsewhere in the text (e.g., p. 4) saturation state is used properly.

-p. 2 lines 22–24 as well as throughout the manuscript: Is it not calcium carbonate which is saturated, but the waters that are saturated *with respect to it*. So, if $\Omega$ is >1 then the waters are oversaturated *with respect to calcium carbonate* and the solid should precipitate. See http://www.biogeosciences.net/10/6453/2013/bg-10-6453-2013.pdf in the same journal for a good definition of carbonate equilibrium.

-p. 4: please give the dissociation constants used (K1 and K2) as well as the Kso used for aragonite and calcite along with references. These values are pivotal to the calculations of saturation states throughout and in the construction of the figures.

- p. 4, l. 30: The first paragraph of the result section is not actually result, but general statements probably more relevant for the discussion. References should be given to substantiate those statements.

-p. 5 *Water undersaturated with respect to aragonite* would be clearer than "low saturated water".

-p.5, l.29 "As is directly evident from Fig. 2 . . . organic matter decay is an important process" . Please discuss which features of Fig. 2 testify of OM degradation.

[Figure]

-p. 6 Figure 3 does not show a correlation (Figure 7 does). Has the correlation between pCO2 and $\Omega$ been quantified?

-Lines 5–10 on p. 6 make a lot of shortcuts in the reasoning leading to the conclusion that "some process must compensate for the CO2 that is consumed by primary productions". What are "low nitrate", "low phosphate" in this context? In general it would be helpful to give the concentrations to better substantiate statements of highs and lows. Further, how are those concentrations evidencing recent primary production? I guess the authors mean that rates of photosynthesis, despite being high as evidenced by depletion of nitrate, were exceeded by respiration, which leads to net accumulation of CO2?

-p. 6 l. 12–13 . There seem to be a problem with the sentence beginning by "As CO2 is supersaturated indicates. . ."

-p. 6 l. 24: Please give a reference (even if it is a textbook one) for the C:N:P:O2 ratio of 106:1:16:-138

-p. 7 l. 9 "it is evident that the aragonite under-saturated water is well confined to the high nutrient of the high Pacific N-P relation signature (Figure 8)." I can see that the samples with $\Omega$ <1 are grouped in the upper-right corner with high N:P ratio, but what and where is the Pacific N-P relation and how does Fig. 8 support that the water masses are exported to the North Atlantic Ocean?

-p. 7 l. 15: What is "the source to the observations of this investigation" ?

-Fig. 7: The variable represented by colour coding is indicated on the upper right corner of each plot, but on plot 7c the variable indicated is PO4 while the caption indicates that the data is colour coded by AOU. Which one is it?

-Fig. 7b: Are those arrows arbitrary ? What determines length, angle, position? This is important since it is used to support the discussion on p. 6 l. 15–18.

-Fig.9: The caption read like a discussion point. Maybe move some of the text to the

discussion on p. 7?

---

## Author Comment (AC1) · 6 Feb 2017

1 We include the suggested reference to Comeau et al (2010).

2 We expand the text with the notation that the HCO3-/H+ ratio or just the H+ are the controlling factors of biogenic calcium carbonate precipitation and include the suggested references.

3 We cite the paper by Haese et al (2014).

4 A sentence describing the consistencies of our measured carbonate system data is included.

5 Finally with regard to the impact of salinity on the solubility of calcium carbonate we are mainly considering its effect on the calcium ion concentration, not on the chemical

equilibrium solubility product or carbonate ion concentration. This is clarified in the next version of the text.
* * *

---

## Author Comment (AC2) · 6 Feb 2017

1 The depth of where the corrosive water spreads in the deep basin is added.

2 Same as comment 5 by reviewer 1.

3 A discussion on the potential importance of photodegradation is included in the discussion and the text describing the river plume and the exchange of $CO_2$ with the atmosphere of this water is altered to clarify the message (that we agree is not clear in the submitted version).

---

## Author Comment (AC3) · 6 Feb 2017

1. As a general action to the comments of lack of references we include such where the reviewer suggests so.

2. We shift the two first paragraphs as suggested.

3. Regarding eq 1-2 the notations "equilibrium" and "observed" are added as a super-script to make this point clear.

4. The expression "solubility state" is changed to "saturation state".

5. We apology for our sloppy expression of "calcium carbonate saturation". All through the manuscript it is changed to "water saturated with respect to calcium carbonate".

6. –p 4. The dissociation constants K1 and K2 are given in the original manuscript, but

we complement with that of the solubility product (Kso according to Mucci, 1983) and salinity-calcium ion concentration ration (Riley and Tongudai, 1967).

7. –p 4, l 30: The intention was to introduce the result section by setting the mind of the reader and then substantiate the statements in the following text. If this is suitable or not we leave to the editor to decide. No problems to change if needed.

8. –p 5. We change from "low saturated water" to "water undersaturated with respect to .."

9. –p 5, l 29. We tried to explain that in the next sentence. As it might not be clear we expand this text to strengthen the arguments.

10. Figure 3. We change "correlated" to "associated" as we don't mean a statistic correlation. No such has been done in the manuscript and we don't feel this add any substantial information.

11. –p5, l 5-10. The text is expanded to make this point clearer to the reader. We take the advice of the reviewer and include some concentrations, as well as parts of the text arguments.

12. –p 6, l 12-13. We do not see the problem here; however, we changed accordingly to clarify the arguments.

13. A reference to Redfied et al., 1963 is included.

14. –p7, l 9. This is what we try to explain in that paragraph, but obviously not well enough. It partly relies on the work of Jones et al (2003) and in the revised version we repeat some of those arguments. Figure 8 does not support export to the North Atlantic by itself, but the signature has been used by Jones et al (2003) to show that some of the Pacific water does. That's why we use the word "likely". Again text is added to clarify the arguments.

15. –p 7, l 15. The source is the Siberian shelf and this is spelled out explicitly in the

next version.

16. Fig 7. Our mistake. Should be PO4 (as also clear from the scaling). This is changed. Thank you for noting.

17. Fig 7b. The only relevance of these arrows is the directions. Length and location is not relevant. This information is added.

18. Fig 9. We feel that it is needed to explain all the details of the figure in the legend so it can be understood by itself. However, we will scrutinize if this text can be optimized.

---

## Author Response (AR1)

BG-2016-478- comments to our revised version

In the following we how we have revised the manuscript based on each of the reviewer's remarks.

Rev. 1.
1.  We include the suggested reference to Comeau et al (2010).
2.  We add a sentence with the notation that the HCO3-/H+ ratio or just the H+ are the controlling factors of biogenic calcium carbonate precipitation and include the suggested references.
3.  We cite the paper by Haese et al (2014).
4.  A sentence describing the consistencies of our measured carbonate system data is included at the end of the method section.
5.  Finally with regard to the impact of salinity on the solubility of calcium carbonate we have changed the text in the beginning of the discussion section to stress that salinity impacts the calcium ion and DIC concentrations (and that pH impacts the chemical speciation of DIC).

Rev. 2.
1.  The depth of where the corrosive water spreads in the deep basin is added.
2.  Same as comment 5 by reviewer 1.
3.  A discussion on the potential importance of photodegradation is included in the discussion and the conclusion sections. The text describing the river plume and the exchange of CO2 with the atmosphere of this water is altered in the conclusion to specify the processes more in detail.
    We also stress that the conclusions are mainly based on our observations that we have discussed in the above section and thus the statements are drawn from that.

Rev. 3.
1.  As a general action to the comments of lack of references we include such where the reviewer suggests so.
2.  We shift the two first paragraphs as suggested.
3.  Regarding eq 1-2 the notations "equilibrium" and "observed" are added as a superscript to make this point clear.
4.  The expression "solubility state" is changed to "saturation state".
5.  We apologize for our sloppy expression of "calcium carbonate saturation". All through the manuscript it is changed to "water saturated with respect to calcium carbonate" or a similar notation.
6.  –p 4. The dissociation constants K1 and K2 are given in the original manuscript, but we complement with that of the solubility product (Kso according to Mucci, 1983) and salinity-calcium ion concentration ratio (Riley and Tongudai, 1967).
7.  –p 4, l 30: The intention was to introduce the result section by setting the mind of the reader and then substantiate the statements in the following text. Now we have moved this part to the introduction.
8.  –p 5. We change from "low saturated water" to "water undersaturated with respect to .."
9.  –p 5, l 29. We changed the text to be more specific to strengthen the arguments.
10. Figure 3. We change "correlated" to "associated" as we don't mean a statistic correlation. No such has been done in the manuscript and we don't feel this add any substantial information.

11. –p6, l 5-10. The text is expanded to make this point clearer to the reader. We take the advice of the reviewer and include specific concentrations, as well as parts of the text arguments.

12. –p 6, l 12-13. We do not see the problem here; however, we changed accordingly to clarify the arguments.

13. A reference to Redfield et al., 1963 is included.

14. –p7, l 9. The text has been changed to i/ specify how Jones et al (2003) defined Pacific originating water, and ii/ include a part of a sentence to couple that Pacific originating signature to the high silicate and low omega.

15. –p 7, l 15. The source is the Siberian shelf and this is spelled out explicitly in the next version.

16. Fig 7. Our mistake. Should be PO4 (as also clear from the scaling). This is changed. Thank you for noting.

17. Fig 7b. The only relevance of these arrows is the directions. Length and location are not relevant. This information is added both in the figure legend and in the discussion text.

18. Fig 9. We have moved the detailed explanations to the main text.